# Primary Mapping and Analysis of the *CmARM14* Candidate Gene for Mature Fruit Abscission in Melon

**Dongyang Dai** [1,2,3]**, Ling Wang** [2]**, Junming Zhang** [2]**, Haojie Qin** [2]**, Huiying Liu** [1,3,*] **and Yunyan Sheng** [2,*]

1   Department of Horticulture, College of Agriculture, Shihezi University, Shihezi 832003, China
2   Department of Horticulture and Landscape Architecture, Heilongjiang Bayi Agricultural University, Daqing 163309, China
3   Key Laboratory of Special Fruits and Vegetables Cultivation Physiology and Germplasm Resources Utilization of Xinjiang Production and Contruction Crops, Shihezi 832003, China
*   Correspondence: hyliuok@aliyun.com (H.L.); shengyunyan@byau.edu.cn (Y.S.); Tel.: +86-138-9952-2503 (H.L.); +86-459-6819-610 (Y.S.)

**Abstract:** Mature fruit abscission (MFA) is an important trait in terms of both harvest and yield. MFA can affect the production and economic value of melon fruit. An $F_3$ population segregating for a single gene and derived from a cross between line M2-10, which shows no mature fruit abscission (None MFA), and the MFA line ZT00091 was used to map candidate genes. Specific length amplified fragment (SLAF) sequencing, in conjunction with bulked-segregant analysis (BSA), was used to map loci governing the natural fruit abscission of plants composing the $F_3$-57 family. A candidate locus, *mfa10.1*, located on chromosome 10 between genomic positions 73,229 and 818,251, was obtained. An insertion-deletion (InDel) marker and 46 recombinant individuals were used to narrow the candidate region to within 35 kb at the genomic position of 650,203 to 685,250; this region included six candidate genes. qRT–PCR gene expression and gene sequence data showed that the *CmARM14* gene, which encodes a RING-type E3 ubiquitin transferase (*MELO3C012406*), was a candidate for melon MFA. Subcellular localization observations revealed that the *CmARM14* fusion protein was localized to the golgi apparatus. Taken together, these results provide a molecular basis for melon breeding.

**Keywords:** melon; mature fruit abscission; gene mapping; *CmARM14*

## 1. Introduction

Fruit abscission is the last developmental stage in the plant life cycle, and many factors involved in fruit abscission are tightly regulated and highly coordinated [1]. During ripening, many fruits undergo fruit abscission, which occurs in the abscission zone (AZ). Mature fruit abscission (MFA) is considered a genetically controlled process during which many extensive transcriptional changes occur [2]. It is difficult to identify the abscission mechanism on the basis of phenotype under various environmental conditions. For agricultural production, the regulatory and stress factors that affect fruit development have substantial consequences for species reproduction. For many crop species, the time of abscission determines the time to harvest and, ultimately, the yield [3]. Therefore, understanding fruit abscission is conducive to allowing mechanized harvesting, reducing human and material inputs, and reducing production costs [4,5].

Melon (*Cucumis melo* L., 2n = 24) is an economically important *Cucurbitaceae* plant species. Because melon fruit can be either climacteric or nonclimacteric, melon has become a model species for research on fruit ripening. Using near-isogenic lines (NILs), Moreno et al. [6] identified the quantitative trait locus (QTL) *eth3.5*, which induces fruit ripening and increases ethylene levels. Afterwards, a new major QTL, *ETHQV6.3*, related to fruit ripening and ethylene production was finely mapped to a 4.5 Mb region [7]. A NAC transcription factor that contributes to climacteric melon fruit ripening has also been identified. To understand the genetic relationship between melon fruit abscission and

ethylene production, Périn et al. (2002) used an $F_2$ population to detect QTLs for fruit ripening, and the results showed that fruit abscission was controlled by two independent genes (*AL-3* and *AL-4*) located on chromosomes 8 and 9, respectively [8]. In 1975, two genes, *AL-1* and *AL-2*, related to fruit abscission were reported in a melon antipowdery mildew study [9]. However, these four loci have not been thoroughly identified or analyzed, so the relationships between them are not clear. In 2020, Pereira et al. [10] conducted a QTL analysis of multiple traits to explain the molecular mechanisms underlying fruit ripening. More than 10 QTLs associated with fruit ripening and abscission were primarily investigated, but the distribution of these QTLs for MFA was relatively scattered.

Using a genetic map, researchers have documented QTLs for abscission in other plant species aside from melon [11–13]. On the basis of a high-density genetic map, researchers found that candidate genes at two major QTLs controlling leaf abscission traits in *Poncirus trifoliata* responded to cold stress treatment [14]. A population comprising 152 soybean recombinant inbred lines (RILs) was used to map the loci governing two different pod abscission rates, and an additive QTL, *cqFARLPG-07*, was detected for flower abscission rates [15]. Similarly, eight QTLs were detected among several *Malus* × *domestica* Borkh linkage groups (LGs), the results of which indicated that fruit abscission represents a self-thinning trait [16]. In addition, using a rice $F_2$ population, researchers studying two QTLs for seed shattering and abscission layer formation have also detected a single QTL named *qSH3* that controls the degree of seed shattering [16]. In 2021, specific length amplified fragment (SLAF)—bulked-segregant analysis (BSA) sequencing technology was applied to a population segregating for a single gene to map one of the candidate loci (*AL3*) for melon MFA, and a 64.7 kb region containing 10 functional genes on chromosome 8 was revealed and further investigated [17].

Here, we primarily mapped a locus, *mfa10.1*, which is located on chromosome 10 and contains a RING-TYPE gene, which is a candidate gene related to melon MFA. We named this gene *CmARM14*, and gene mapping and candidate gene analysis with subcellular localization were subsequently conducted to understand melon MFA.

## 2. Materials and Methods

### 2.1. Plant Materials and Inheritance Analysis

M2-10, a line whose fruit is thin skinned and for which no abscission occurs when the fruit has ripened (~32 days are needed to reach maturity after pollination), and ZT00091, which was developed at the Zhengzhou Fruit Research Institute, a line whose fruit is thin skinned, for which MFA occurs at 25 days after pollination (DAP) and for whose fruit ripen at approximately 30 DAP, were used. The $F_1$ was obtained using a hybrid combination of M2-10 and ZT00091. The $F_1$ was backcrossed to the parents to obtain $BC_1P_1$ and $BC_1P_2$, respectively (Figure 1A). One hundred forty-four $F_2$ plants as well as $F_{2:3}$ families were derived from a cross of M2-10 × ZT00091 melon lines for the identification of MFA characteristics; 150 individuals of the $F_{3:4}$-57 population segregating for the single *AL4* gene were used for gene mapping in the Heilongjiang Bayi Agricultural University greenhouse in 2017 and 2018. To determine the inheritance characteristics of MFA, one hundred fifty $BC_1P_1$ and one hundred thirty-four $BC_1P_2$ plants were planted in the field in 2019.

Seven natural populations were added and used to test the accuracy of the markers in the Heilongjiang Bayi Agricultural University greenhouse in 2022. The seven natural populations were 1244, MS5, WI998, P5, P10, S8, and S7.

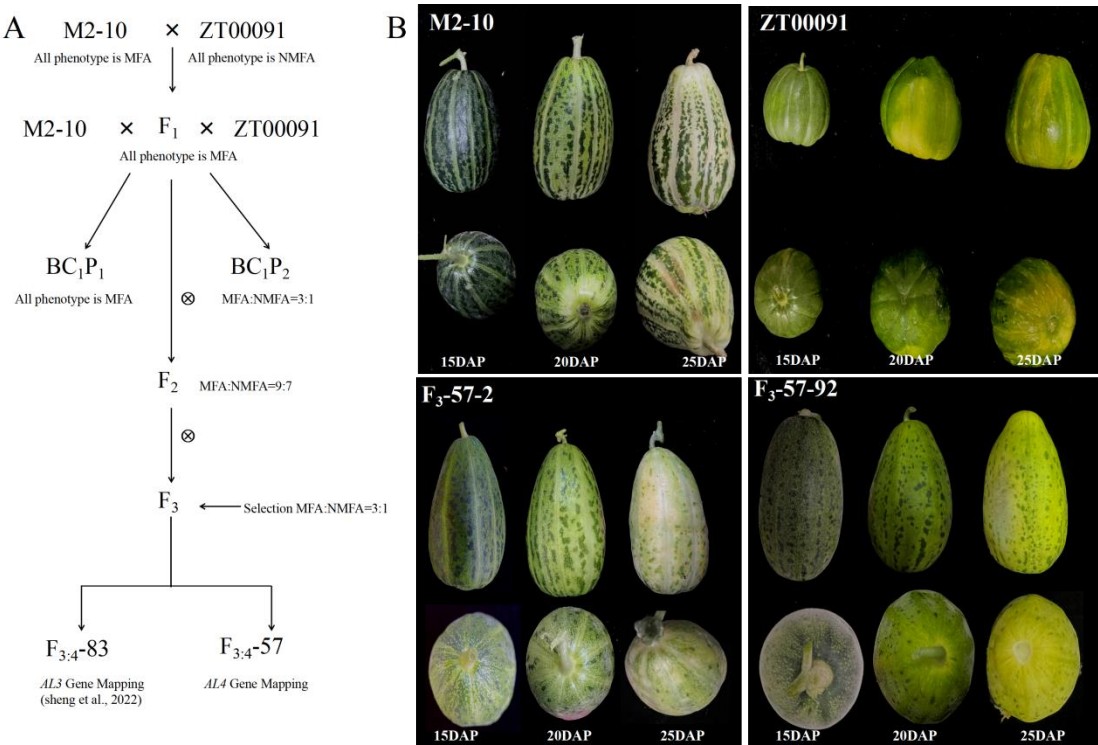

**Figure 1.** Technology roadmap and phenotype of M2-10 and ZT00091. (**A**): Technology roadmap of this study. (**B**): Phenotype of M2-10 and ZT00091. Each material had three fruits from 15 DAP (days after pollination), 20 DAP, and 25 DAP. For M2-10, fruit matured about 32 days after pollination and without abscission until ripen. Fruits of ZT00091 became mature from 28 DAP but started to abscise from ~25 days. $F_3$-57-2 and $F_3$-57-92 were the $F_3$ family of No.57 from M2-10 crossing with ZT00091, the fruit of which were without mature abscission and with mature non-abscission, respectively.

### 2.2. Identification of MFA

For the investigation of mature fruit abscission, the mature fruits from each plant were investigated from 25 to 30 days after pollination. For families, fruits from all plants were detected and harvested, and ripening behavior was classified, with "A" representing all plants whose mature fruit did not undergo abscission, "B" representing all plants whose mature fruit underwent abscission, and "H" representing plants whose mature fruit showed both phenotypes. Analysis of the chi-square test ($X^2$) was performed using SAS software (SAS Institute Inc., Cary, NC, USA).

### 2.3. Construction of Bulks for Preliminary Mapping

For sequencing of the bulks, twenty $F_2$ populations exhibiting no MFA segregation were selected for self-pollination, and twenty $F_3$ families were obtained. All the $F_3$ families were planted in spring 2018 at the Heilongjiang Bayi Agriculture University Experimental Station, and each family comprised ~150 plants. After investigating the segregation of MFA/mature fruit non-abscission (None MFA) of the twenty $F_3$ families, Nos. 83, 46, 57, 97, and 110 were found to fit a 3:1 segregation ratio (MFA:None MFA), and the plants composing these five $F_3$ families were self-pollinated. After performing a linkage marker analysis for *CmAL3* [17], we found that the $F_3$-57 family represented a population that segregated for the single gene; consequently, this family was selected as a material for bulk construction for mapping the *AL4* candidate gene.

Thirty homozygous MFA and None MFA plants from the $F_3$-57 population were selected based on the phenotype of the plants composing the $F_4$-57 generation (30 plants per family) and used to construct bulks for SLAF sequencing.

### 2.4. Whole-Genome Sequencing of Parental Lines and Bulk SLAF-Seq

The two bulks and the M2-10 and ZT00091 parental lines were subjected to sequencing using an Illumina GAIIx machine. The preparation of the genomic DNA from the parental lines and the two bulks, as well as the selection of the PCR settings, was performed according to the Illumina sample preparation guide. The products were purified at appropriate sizes (260–420 bp) and then excised and diluted for sequencing by an Illumina GAIIx (Illumina, Biomark, Beijing, China). The clean read sequences were compared to the reference genome (http://cucurbitgenomics.org/) (accessed on 8 May 2021). The allele frequency was calculated by the Euclidean distance (ED), followed by a Loess regression analysis, which can identify a region in which the gene governing a target trait lies and generate a list of putative regions within the linked genomic segment.

### 2.5. Fine Mapping

Based on the resequencing data of the parental lines, single-nucleotide polymorphisms (SNPs) were analyzed, and 11 insertion–deletion (InDel) markers and 3 simple sequence repeat (SSR) markers were used to finely map the *CmARM14* gene in the candidate region (Table 1). Additionally, 352 $F_3$-57 plants and the $F_4$-57 families (each containing 30 plants) were used to fine-map the candidate genes. Plants were also genotyped by polymorphic markers to determine the candidate position of the locus.

**Table 1.** Primer information used for fine mapping.

| ID | Marker Name | Chromosome | Genome Position | Forward Sequence | Reverse Sequence |
|---|---|---|---|---|---|
| 1 | Indel1-1 | 10 | 194,740 … 195,031 | ACGTTGCAAGTGTGCCTTATC | GTGAGTGAGGTAGGTGTGGA |
| 2 | Indel2 | 10 | 234,249 … 234,583 | TTTAGGTTTGTGAGGAAGAAGG | TCCCCCCAATAAAGTAAAACA |
| 3 | CmSSR23348 | 10 | 234,705 … 235,730 | CATTGATCCAAATGTTACCCAA | GGGAGGGGCTATGGGTTAT |
| 4 | Indel27 | 10 | 563,304 … 563,535 | TGGCCTTTTTGACTGCTCTT | TAAATCTCCTACCGTTCCCG |
| 5 | Indel28 | 10 | 575,734 … 575,971 | AAGAATGTATGGATTAAAAGGGTTT | TCTTTCGCTCATGGAAGTCA |
| 6 | SNP240 | 10 | 579,866 … 580,121 | AACTCCTTCCTTTTCCCGTC | GCATCAAAACCATTTTTCTTTG |
| 7 | SNP255 | 10 | 608,958 … 609,170 | AGTTTTGTCAACCACACCCA | TTTCTCCGATTTCTTTCCATTC |
| 8 | Indel31 | 10 | 616,081 … 616,308 | TCGAAGATGGTCCATTAGGG | TTGGAAGCAAAGCAACTCCT |
| 9 | SNP271 | 10 | 650,203 … 650,387 | CCCTCCCTCCCTCTAAATGA | GGATGTTGCCTTGAAAAAGC |
| 10 | SNP286 | 10 | 685,250 … 685,509 | CCGTCGCGTTCGTTTTATTA | AAAAATGAAATTGGCAGCACTT |
| 11 | SNP301 | 10 | 719,646 … 719,884 | AACCCCTCAATAACCCAACC | GAGTGAAGCAAACACCACGA |
| 12 | SNP316 | 10 | 738,058 … 738,298 | GGCCCCATAATTGAGAAAAA | ATTGCATGCATGTGGATGAT |
| 13 | CmSSR23418 | 10 | 742,790 … 743,815 | ATTTGATTTTTGCAAAGCGG | CAAAATTGGGGAGTATAGTGACAA |
| 14 | CmSSR23419 | 10 | 761,037 … 762,060 | TTGGTAGGCTAAAAGATCGTCC | CAAAAGACACATCATGAGGGC |

Genomic DNA was extracted from young leaves of each individual plant via the cetyl-trimethylammonium bromide (CTAB) method [18]. Each bulk was made by mixing equal amounts of DNA from 20 homogeneous plants with MFA and plants with None MFA. DNA quality and concentration were measured via 1% agarose gel electrophoresis, and the final DNA concentration was adjusted to 75~100 ng/μL.

Each PCR mixture included 30 ng of template DNA, forward and reverse primers (1.0 μM each), a mixture of dNTPs (0.2 mM), 0.1 unit of Taq DNA polymerase, and 1× PCR buffer (Takara, Beijing, China); the total volume was 10 μL. A 6% polyacrylamide gel with silver staining was used to separate the digested products. JoinMap 4.0 was used to determine the LGs for association regions, and the Kosambi map function was used to calculate the genetic mapping distance between markers.

### 2.6. qRT–PCR

Total RNA was isolated from the pedicels of individuals in the parental line to determine candidate gene expression. Pedicels at different stages of fruit development (15, 20, and 25 DAP) were selected for RNA isolation via RNAiso Plus (Takara, Beijing, China). qRT–PCR was performed using an IQ5 system (Bio-Rad, Hercules, CA, USA); 10 μL was used. The PCR primers used were designed using Primer 5.0 software. The sequences of the candidate genes and their primers are shown in Table S2. Each PCR mixture consisted of 10 μL, which included 2× TranStart Top Green qPCR SuperMix (TransGen, Beijing, China), 10 pmol of each primer, and 1.5 μL of cDNA template, and ddH$_2$O was added such that the final volume was 10 μL. qRT–PCR was performed as follows: 95 °C

for 15 s, 55 °C for 15 s, and then a slow increase in temperature (by 0.5 °C per cycle) to 95 °C, during which time the fluorescence was continuously measured. Three biological and technical replicates were subjected to qRT–PCR, and the gene expression data were collected using the ΔCt method.

### 2.7. Subcellular Localization

A cDNA template gene in conjunction with specific primers was used to amplify the full-length *CmARM14* according to the following protocol: denaturation at 94 °C for 4 min; 45 cycles of denaturation at 94 °C for 30 s, annealing at 50~64 °C for 30 s, and extension at 68 °C for 1 min/kb; and maintenance at 68 °C for 8 min. Then, the PCR products were reconstituted with the vector, and the recombinant products were transformed into living *Escherichia coli* competent cells. The transformed bacteria were subsequently coated onto LB solid media supplemented with kanamycin (50 mg/L) and cultured overnight for PCR verification, after which positive clones were selected for sequencing. The recombinant plasmids were extracted and transferred into Agrobacterium using the heat shock method. Afterwards, tobacco leaves were infected via the Agrobacterium-mediated method, and subcellular localization of the *CmARM14* protein was observed. An FV1000 laser-scanning confocal microscope (LSCM IX83-FV1200) was used for the observations. The relevant primer information is shown in Table S2.

### 2.8. Candidate Gene Analysis

The DNA sequences were retrieved from the NCBI database (http://blast.ncbi.nlm.nih.gov/Blast.cgi/) (accessed on 17 May 2022) and by the use of the cucurbit dataset (http://cucurbitgenomics.org/) (accessed on 20 May 2022) to find the genes contained in the candidate regions. The structure of the genes was analyzed via FGENESH (http://linux1.softberry.com) (accessed on 20 May 2022), and MEGA 6.0 software was used to analyze genetic evolution. Analysis of candidate gene structural domains and prediction of interacting proteins was performed using the online site SMART (http://smart.embl-heidelberg.de/smart/set_mode.cgi?GENOMIC=1) (accessed on 2 June 2022).

## 3. Results

### 3.1. The $F_3$-57 Family Constitutes a Population Whose Members Segregate for Single Gene at the mfa10.1 Locus

M2-10 and ZT00091 are different subspecies of melon whose fruit are thin skinned. The M2-10 fruit matures at approximately 32 DAP, and there is abscission until ripening. ZT00091 fruit matures at approximately 28 DAP, and abscission begins at 25 DAP (Figure 1B). $F_1$ progeny resulting from a cross between M2-10 and ZT00091 revealed MFA, and the segregation ratio among the $F_2$ individuals was 9:7 (None MFA:MFA), indicating that MFA was controlled by two complementary genes ($X^2$ = 0.173). With respect to identifying a population that segregated for a single gene, five $F_3$ families and the $F_4$ population were used to determine the segregation ratios, and $X^2$ goodness-of-fit tests indicated that the segregation ratios of the $F_3$-57 ($X^2$ = 0.33), $F_3$-83 ($X^2$ = 0.11), and $F_3$-97 ($X^2$ = 1.11) populations were consistent with the expected 3:1 ratio (Table S1). However, a previous study revealed that MFA in the $F_3$-83 and $F_3$-97 families was controlled by *AL3* genes, and gene mapping was also conducted (Sheng et al., 2022). Subsequent marker-assisted selection and phenotypic performance indicated that MFA in $F_3$-57 was controlled by the *mfa10.1* locus (Figure 2).

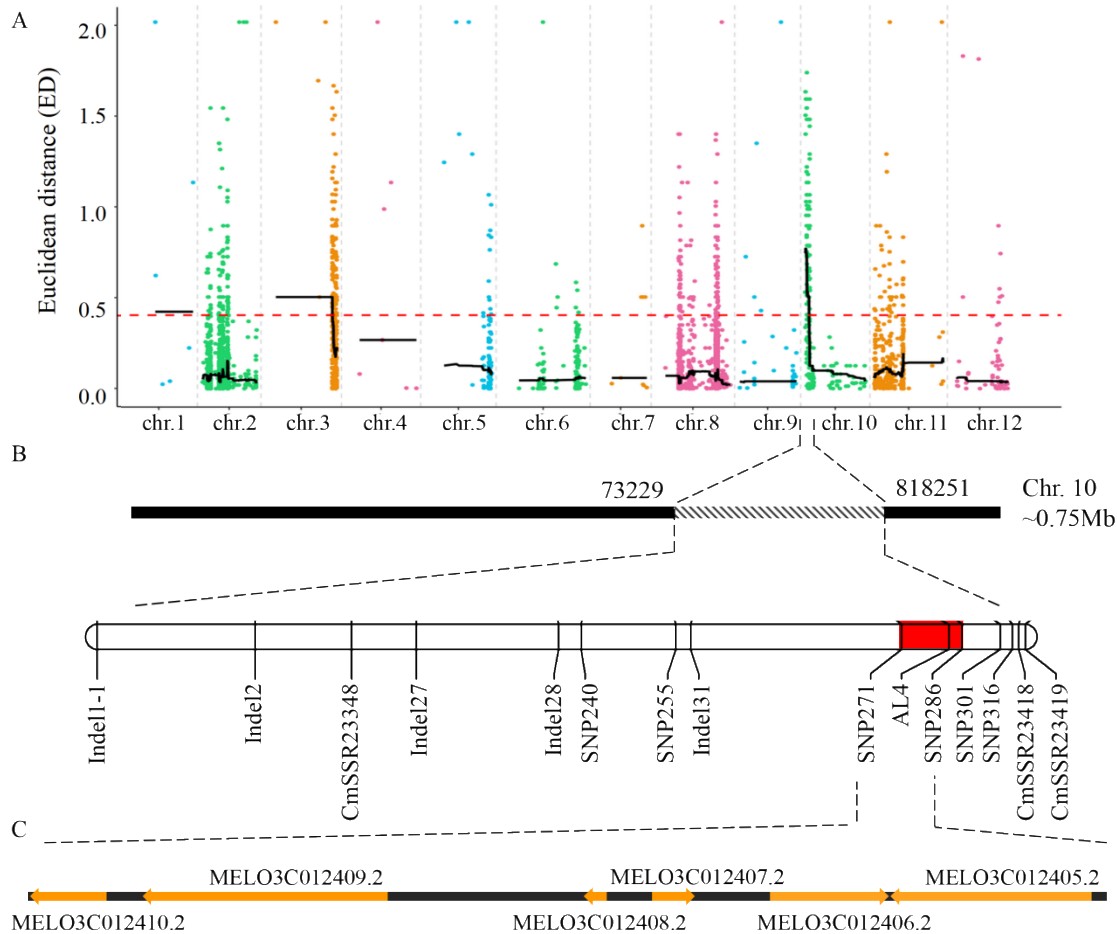

**Figure 2.** Mapping of candidate genes for MAF. (**A**): Primarily mapping of *mfa10.1* using SLAF-BSA technology by F$_3$-57 family, which is a single gene segregation population. (**B**): narrow mapping of the candidate region in chromosome 10, and more than 14 molecular markers were used to narrow the candidate region by another 352 F$_3$-57 individuals plants. (**C**): Finally, the candidate gene was located between SNP271 and SNP286, which spanned the 35 Kb region containing contains 6 candidate genes.

### 3.2. Primary Mapping of the mfa10.1 Locus by the SLAF-BSA Method

The None MFA and MFA bulks were constructed by mixing equal amounts of DNA from 30 individuals with and without MFA from the F$_3$-57 population. SLAF-BSA sequencing was conducted for the primary mapping of *mfa10.1*. For the M2-10 and ZT00091 parental lines, an average of 44.07 Gb of clean data (with a Q30 of 92.95%) was obtained, and the sequencing data of the clean data for bulks amounted to 45.46 and 42.68 Gb, respectively. The Q30 was 92.88% and 93.03%, and approximately 87.53% and 89.48% of the reads mapped to the reference genome (Table 2).

**Table 2.** Sequencing data for parental lines and bulks.

|  | Clean Reads | Clean Base | Q30 (%) | GC (%) | Total Reads | Mapped (%) | Properly Mapped (%) |
|---|---|---|---|---|---|---|---|
| ZT00091 | 45,492,988 | 13,622,302,478 | 92.88 | 38.03 | 90,985,976 | 95.04 | 87.53 |
| M2-10 | 42,680,203 | 12,780,598,726 | 93.03 | 36.94 | 85,360,406 | 97.18 | 89.48 |
| MFA BULK | 36,547,360 | 10,943,443,284 | 91.68 | 36.13 | 73,094,720 | 98.77 | 93.12 |
| None MFA BULK | 31,846,911 | 9,529,678,748 | 93.74 | 37.01 | 63,693,822 | 96.78 | 90.76 |

A total of 172,910 SNPs, including 3603 nonsynonymous SNPs, were identified in the parents. For the bulks, a total of 87,155 SNPs, 1800 of which were nonsynonymous, and 30,952 small InDels between the bulks were identified.

After filtering and analyzing the SNPs identified from the parental analysis, we analyzed 14,090 SNPs obtained from the two bulks of the $F_3$ generation for trait association using the ED and ΔSNP-index calculation methods. Based on the ED method, when the correlation threshold was 0.34, *mfa10.1* was detected in the melon genome between positions 23,044 and 2,056,273 on chromosome 10. However, when the ΔSNP-index method with a correlation threshold of 0.53 was used, *mfa10.1* was detected in the melon genome between positions 23,044 and 1,558,140 on chromosome 10. Combining these two results, we found that the intersection of the associated regions for *mfa10.1* was located on chromosome 10 at the genomic position between 23,044 and 1,558,140 (Table 3).

**Table 3.** The linkage region using by two calculations with 2 analysis methods.

| | Chromosome | ED Analysis Method | | Index Analysis Method | | Combination | | Gene Number |
| | | Genome Start | Genome End | Genome Start | Genome End | Genome Start | Genome End | |
|---|---|---|---|---|---|---|---|---|
| SNP methods | 10 | 23,044 | 2,057,566 | 23,044 | 1,558,140 | 23,044 | 1,558,140 | 206 |
| INDEL methods | 10 | 73,229 | 1,779,001 | 73,229 | 818,251 | 73,229 | 818,251 | 111 |
| Combination region | 10 | | | | | 73,229 | 818,251 | 111 |

In total, 3295 InDels were used for trait association by the ED and ΔInDel-index methods after filtering and analysis. Based on the ED calculation method, with a threshold of 0.40, *mfa10.1* was mapped to chromosome 10 at a genomic position of 73,229 to 11,779,001. For the InDel-index calculation method with a threshold of 0.53, *mfa10.1* was mapped to chromosome 10 at a genomic position of 73,229 to 818,251. Combining these two results, we found that the intersection of the associated regions was located on melon chromosome 10 at a genomic position of 73,229 to 818,251 (Table 3).

Comparing SNP and InDel analysis with ED and index calculation methods, we found that the primary mapping results indicated that the MFA locus (*mfa10.1*) was located on melon chromosome 10 and that the genomic position was between 73,229 and 818,251. This region corresponds to a physical distance of approximately 750 kb on Scaffold000016 and encompasses 111 genes, 101 of which are annotated in the Melon (DHL92) 3.6.1 genome database (http://cucurbitgenomics.org/organism/18) (accessed on 8 May 2021).

*3.3. Fine Mapping of mfa10.1 and Analysis of the Candidate Gene CmARM14*

To narrow the *AL4* gene candidate region, 46 recombinants from 352 $F_3$-57 plants were used to finely map the *mfa10.1* locus. The progeny of the recombinants ($F_4$ progeny resulting from the self-crossing of the forty-six recombinants) were planted (~20 plants from each recombinant family) in an open field, and their phenotypes were evaluated. Fourteen markers, including SNP, InDel, and SSR markers, were developed by comparing the physical genome sequences of the intervals at the candidate regions between M2-10 and ZT00091 (Figure 2). Seven natural populations were used to test the accuracy of the markers. The mature fruit non-abscission gene maker was identified with 100% accuracy (Figure S1). The *AL4* gene candidate region was 35 kb, spanning positions 650,203 to 685,250, and included six candidate genes. The annotated candidate genes included thioredoxin-like 3-2, chloroplastic isoform X8 (*MELO3C012410*), amino acid transporter family protein (*MELO3C012409*), unknown protein (*MELO3C012408*), SKP1-like protein 12 (*MELO3C012407*), RING-type E3 ubiquitin transferase (*MELO3C012406*), and vacuolar protein sorting-associated protein 33-like protein (*MELO3C012405*) (Table 4). To determine gene expression patterns, qRT–PCR was conducted on the fruit peduncle and pedicel zone tissues at different stages—15 DAP, 20 DAP, and 25 DAP—between the parents. For *MELO3C012405*, there was no difference in expression levels between the parents at any of the three stages, and the gene expression increased and then decreased. For *MELO3C012406*

expression, significant differences were detected at all three stages between the parents; for M2-10, the gene expression level decreased, but the opposite was true for ZT00091. The *MELO3C012406* gene was expressed almost six times more at 25 DAP than at 15 DAP in ZT00091. For *MELO3C012407*, gene expression moderately increased at 20 DAP in ZT00091, and a significant difference was observed at this stage between the parents; however, no difference in the expression level of this gene was found between 15 DAP and 25 DAP. Although a significant difference in expression was detected for *MELO3C012407* at the three stages, with a sharp increase in the gene expression level in M2-10, the expression pattern of *MELO3C012407* in ZT00091 was more complex. Compared with those in M2-10, the *MELO3C012408* gene expression levels in ZT00091 at 15 DAP and 20 DAP were moderately high, but at 25 DAP, they sharply decreased. The expression patterns of *MELO3C012409* and *MELO3C012410* were significantly different at 15 DAP, but there was no difference between the parental lines at 20 DAP and 25 DAP (Figure 3). Based on the gene expression patterns, we found that two genes, *MELO3C012406* (a RING-type E3 ubiquitin transferase gene) and *MELO3C012408* (which encodes an unknown protein), were differentially expressed; therefore, we suspected that these two are candidates for the *AL4* gene.

**Table 4.** Predicted genes between markers.

| Melon Gene ID | Gene Annotation | Physical Location |
| --- | --- | --- |
| MELO3C012410.2 | thioredoxin-like 3-2, chloroplastic isoform X8 | 650,442 . . . 652,744 |
| MELO3C012409.2 | Amino acid transporter family protein | 654,175 . . . 661,881 |
| MELO3C012408.2 | Unknown protein | 668,489 . . . 669,000 |
| MELO3C012407.2 | SKP1-like protein 12 | 670,514 . . . 671,673 |
| MELO3C012406.2 | RING-type E3 ubiquitin transferase | 674,353 . . . 677,946 |
| MELO3C012405.2 | Vacuolar protein-sorting-associated protein 33-like protein | 676,286 . . . 685,240 |

Then, the structure and protein sequence of these two genes from the parent resequencing data and the reference genome sequence data were analyzed. The results indicated that there were no differences in *MELO3C012408* among the M2-10, ZT00091, and reference gene sequence data from the Cucurbit Genomics Database. However, for *MELO3C012406*, an SNP was detected between its sequence at the reference genome sequence; specifically, there was a "T" to "A" inversion in the coding DNA region, which induced a change from amino acid "N" to "K" (Figure 4A). To understand the genetic evolutionary relationship of the *MELO3C012406* candidate gene, 12 related genes from various Cucurbitaceae plant species were used for analysis, including the following: *KAG6573848*, *KAG7012913* (*Cucurbita argyrosperma* subsp. *sororia*), *XP022945252* (*Cucurbita argyrosperma* subsp. *argyrosperma*), *XP023541913* (*Cucurbita moschata*), *XP002967020* (*Cucurbita pepo* subsp. *pepo*), *XP022139309* (*Cucurbita maxima*), *XP0038891943* (*Momordica charantia*), *XP004142402* (*Benincasa hispida*), *XP008446840* (*Cucumis sativus*), *TYK09254* (*Cucumis melo*), *XP022956820* (*Cucumis melo* var. *makuwa* and *Cucurbita moschata*), and *KAG7031978* (*Cucurbita argyrosperma* subsp. *argyrosperma*). The E3 ubiquitin transferase gene *MELO3C012406* is highly homologous to XP008446840 of melon (*Cucumis melo* L.) but is distantly related to the homologue in pumpkin (Figure 4B).

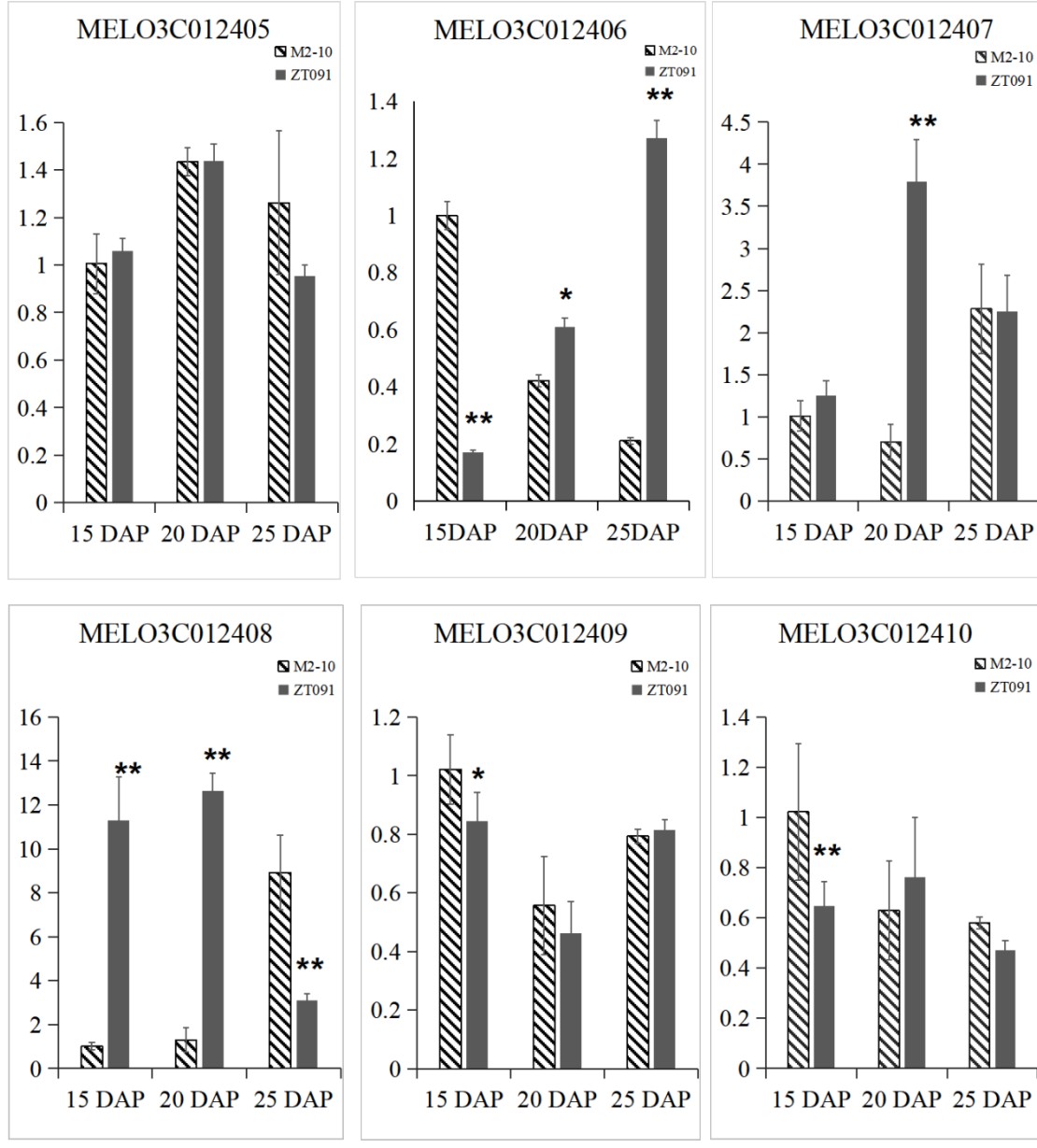

**Figure 3.** Relative expression levels of six candidate genes in None MFA melon 'M2-10' and MFA melon 'ZT00091'. Six genes were quantified using the $2^{-\Delta\Delta CT}$ method. For the two parents, the expression level of the respective genes was determined at 15, 20, and 25 days after pollination, respectively. Each was repeated three times, and 5 plants were mixed in equal amounts to form one replicate in two parents. The striped bar represents M2-10, and the dark gray bar represents ZT00091. ** indicates an extremely significant difference, $p < 0.01$; * indicates a significant difference, $p < 0.05$.

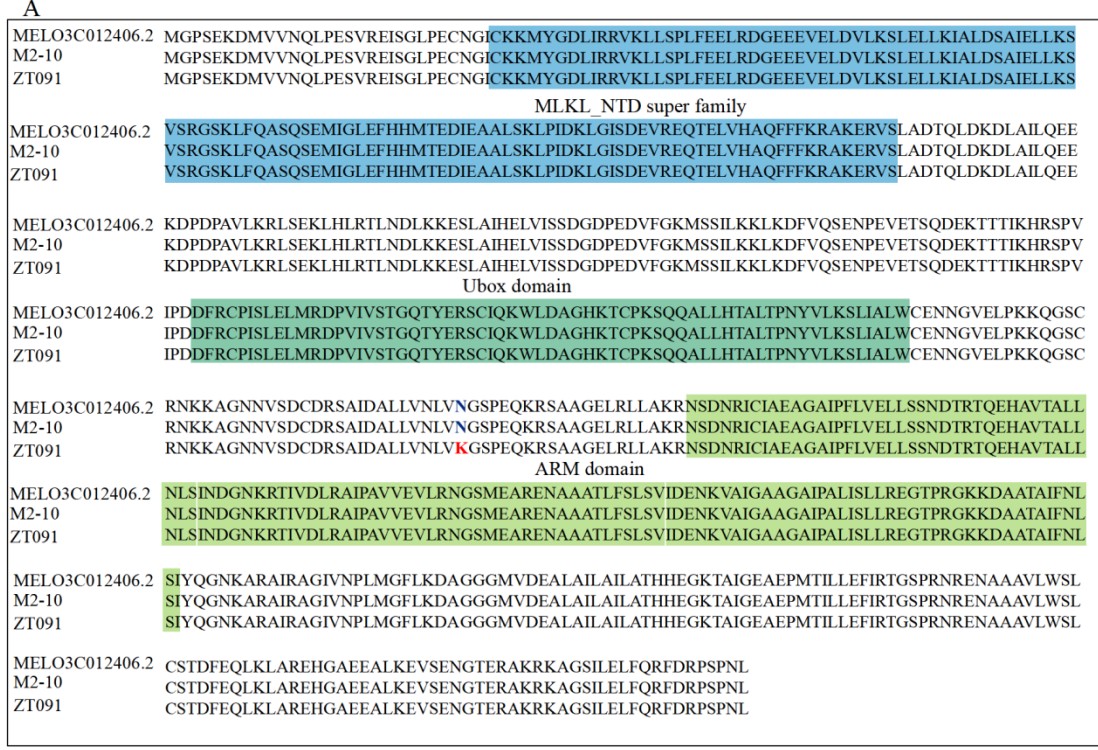

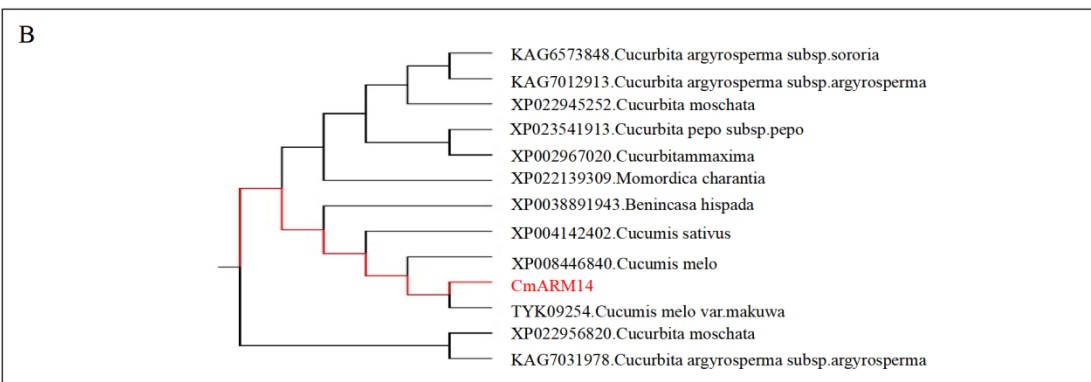

**Figure 4.** Phylogenetic analysis of the MELO3C012406 gene. (**A**): Alignment of protein sequences of reference gene sequences, M2-10 and ZT00091, and the MELO3C012406 gene. The colors are labeled as the structural domains contained in the gene. (**B**): Phylogenetic analysis with other Cucurbitacea plants viz., *Cucurbita argyrosperma* subsp. sororia, *Cucurbita argyrosperma* subsp. argyrosperma, *Cucurbita moschata*, *Cucurbita pepo* subsp. pepo, *Cucurbita mmaxima*, *Benincasa hispada*, *Cucumis sativus*, *Cucumis melo*, *Cucumis melo* var. makuwa, and *Cucurbita moschata*.

### 3.4. Subcellular Localization of CmARM14

The results of the gene structure analysis indicated a RING finger domain, a U-box dominant profile, and an ARE-repeat. Usually, E3 ligases are localized in the nucleus or cytoplasm (Sun et al., 2019). To determine the subcellular localization of *CmARM14*, a *pCmARM41:CmARM14-GFP* construct was introduced into tobacco leaves and evaluated using a confocal microscope. The results revealed that *CmARM14* was present in the Golgi apparatus of the *Nicotiana benthamiana* leaf cells (Figure 5).

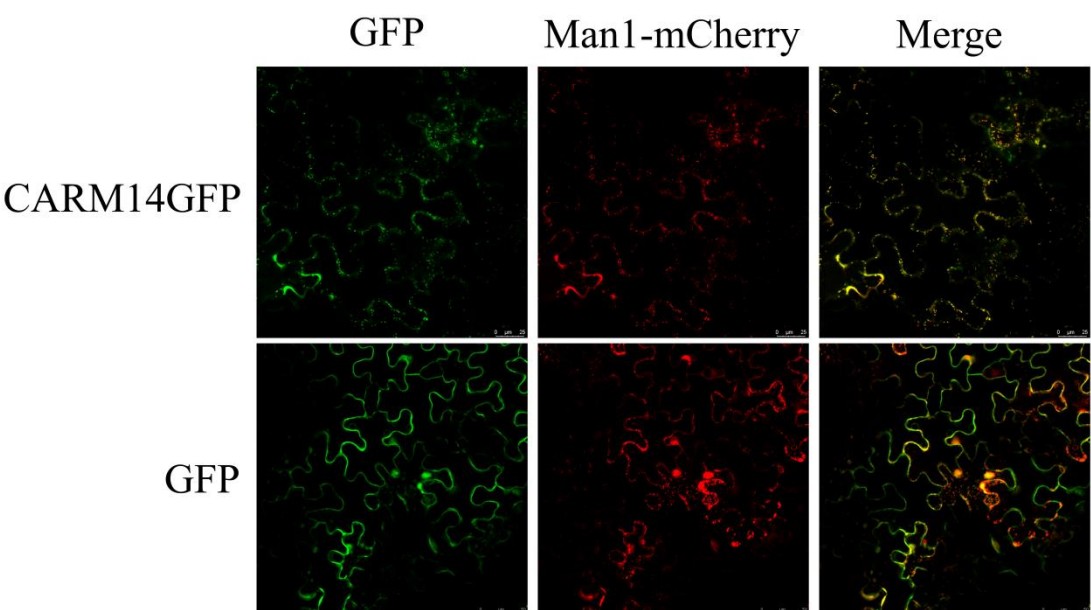

**Figure 5.** Subcellular localization of the *CmARM14* protein. The CmARM14GFP fusion construct and *GFP* gene driven by the *CaMV-35S* promoter were transiently expressed in *Nicotiana benthamiana* individually. The figure shows that the *CmARM14* fusion protein was completely colocalized with the Golgi marker (Man1) protein. CmARM14GFP Scale bar = 25 μm. GFP Scale bar = 50 μm.

To understand whether *CmARM14* regulates fruit abscission, interaction proteins were predicted, and the results revealed that BRASSINOSTEROID INSENSITIVE 1-associated receptor kinase 1-like isoform X1 (BR1, XP_008455226.1, XP_008440121.1), RING-type E3 ubiquitin transferase (XP_008446840.1), the protein kinase PINOID2 (XP_008457762.1), chitin elicitor receptor kinase 1 (XP_008455461.1), and six other proteins that could interact with it (Figure S2).

## 4. Discussion

MFA is a key characteristic of fruit ripening; MFA is not only related to the yield and transportation of horticultural fruit but is also important for maintaining moderate quality traits and minimizing postharvest deterioration [19]. Based on previous research, duplicated independent loci (*AL3* and *AL4*) were found to control melon MFA [20]. More recently, a candidate region for *AL3* identified via SLAF-BSA was mapped to chromosome 8, and six candidate genes were identified [17]. In our research, a population segregating for a single gene was selected to determine the candidate region of the *AL4* locus, and a major locus, *mfa10.1,* was identified in the melon genome between positions 23,044 and 2,057,566 on chromosome 10. Further mapping using InDel markers revealed six candidate genes between positions 650,442 and 685,240. QTLs on chromosome 10 were not the first QTLs identified as being involved in melon ripening, as *ff10.2* was shown to govern climacteric behavior due to an increase in fruit flesh firmness [6]. Enzymes involved in cell wall degradation, including members of the xyloglucan endotransglycolase/hydrolase (XTH) gene family and expansins, were also identified using the candidate gene approach. More recently, a QTL of fruit firmness, *FIRQV10.1*, was found to be located near chromosome 10 at a genomic position of 1,736,076, and based on the interaction of each QTL, there is likely a single QTL for fruit ripening in the interval between 48,430 and 1,736,076 bp [10]. When our results are combined with those of previous research, it becomes apparent that QTLs on chromosome 10 are important for melon fruit ripening. For MFA, using a non-climacteric accession, PI 161375, and the climacteric Védrantais line (*Cantalupensis*), Périn et al. (2002) revealed the presence of two genes (*Al3* and *Al4*) involved in ethylene-dependent fruit abscission, and recent research has identified these two genes responsible for abscission

layer formation on chromosomes 8 and 9, respectively [8,10]. These results were not the same as ours, indicating that many genes involving complex mechanisms are involved in fruit ripening and abscission.

For *mfa10.1*, there were six annotated genes in the candidate region. On the basis of the qRT–PCR results, only two genes, one encoding a RING-type E3 ubiquitin transferase (*MELO3C012406*) and an unknown gene (*MELO3C012408*), were significantly differentially expressed in the three peduncle and pedicel AZs of the fruit. To further determine the candidate gene, gene structure differences between the parental lines were examined (Figure 4A), the results of which suggested that an inversion could be responsible for changes in gene function. Our gene cluster analysis of select homologous genes within Cucurbitaceae plant families also showed that the *TYK09254* gene (which encodes U-box domain-containing protein 14) (Figure 4A) is closely related to our candidate gene (Figure 4B). Interestingly, the results of our analysis of the conserved structural and functional domains of this protein indicated that the N-terminal domain of the mixed lineage kinase domain-like protein (MLKL), cellulose synthase-interactive protein, and modified RING finger domain structures were present in this protein, and the gene structure revealed a U-box dominant profile and the presence of an ARM repeat. Above all, we named the candidate gene *CmARM14*, which is an annotated RING-type E3 ubiquitin ligase-related gene.

E3 ubiquitin ligase plays an important role in plants. RING-type E3 proteins have a specific ring finger domain, and many studies have found that this family of genes plays a key role in regulating protein structure and activity, as well as in regulating plant growth and development and responses to biotic and abiotic stresses [21–24]. Among ubiquitin ligases, E3-type ligases are responsible for recognizing the substrate in the ubiquitination pathway and play important roles in the process of substrate degradation, but the functional mechanisms of only approximately 1% of E3 ligases have been identified. In plants, E3 ligases are involved mainly in the regulation of the cell cycle, hormone levels, seed germination, root development, chloroplast development, response to high-salt conditions, and drought [25].

E3 ubiquitin ligase is an enzyme responsible for targeting proteins to the degradation pathway and plays a major role in a variety of biological activities [22,26]. Generally, the E3 ubiquitin ligase family of proteins includes HECT-type, RING-type, U-box type, and SCF complex types, all of which play important roles in plant growth and developmental processes [27–29]. Many E3 ubiquitin ligases have been identified as playing key roles in fruit ripening. A RING-type E3 ubiquitin ligase named *MaLUL2* was isolated and characterized from banana fruit [19]. In peach, the *PpE3* gene was predicted to be involved in ethylene, auxin, or abscisic acid (ABA) synthesis, the hormones of which are suspected to play a role in peach flesh and fruit ripening [26]. In Arabidopsis, *CONSTANS* interacts with the E3 ubiquitin ligase *COI1* (*COP1*) indirectly and is involved in the jasmonate signaling pathway; thus, *CONSTANS* is involved in the flower organ abscission process [30]. Recently, knockdown of the E3 ubiquitin ligase family gene *SP1* or its homologous gene *SPL2* was shown to delay tomato fruit ripening, while overexpression of *SP1* accelerated fruit maturation [23]. In melon, the RING-type E3 ubiquitin ligase family gene *CmRMA1H1* was found to regulate plant hormone signal transduction for fruit ripening [31]. Above all, E3 genes play important roles in fruit ripening. MFA is also part of the ripening process, but RING-type E3 ubiquitin ligases and their functions have not been characterized in horticultural plant species, especially melon.

In the present study, a candidate gene annotated as an E3 ubiquitin ligase in melon was identified as being involved in MFA. Previous research on another gene, *CmRMA1H1* [31], which belongs to the RING-type E3 ubiquitin ligase family, revealed that this gene could promote fruit softening after ripening. The results suggest that the E3 ubiquitin ligase family genes have an important function in melon fruit ripening, although a previous study also focused on the functions of these genes in the stress response [32]. The transcriptomic profile for MFA indicated that the E3 ubiquitin–protein ligase ABI3-interacting protein 2 (AIP2) was significantly differentially expressed at 38 DAP, and the expression

level was almost 7 times that at 36 DAP and 40 DAP [20]. The results also showed that E3 ubiquitin–protein ligase is active in melon. In *Solanum lycopersicum*, knockdown of the plastid ubiquitin E3 ligase *SP1* or its homologue *SPL2* delayed tomato fruit ripening [23], and in *Vitis*, *VlPUB38* was found to mediate ABA synthesis and its involvement in the regulation of fruit maturation [33]. More recently, two RING finger E3 ligases, *MaBRG2/3*, were found to interact with and ubiquitinate *MaMYB4*, contributing to the ripening of *Musa paradisiaca* fruit.

Our study confirmed that an E3 ubiquitin ligase (*CmARM14*, *MELO3C012406*) plays a negative role during MFA in the development of the peduncle and the pedicel AZ of melon. The expression of the *CmARM14* gene differed significantly across the three developmental stages, and for M2-10, the gene expression level decreased, but for ZT00091, it increased. For each stage (15 DAP, 20 DAP, and 25 DAP), *CmARM14* showed significant expression levels, indicating that this gene is involved in melon fruit development. In recent years, research has indicated that E3 ubiquitin ligase proteins are considered to have a key function in the fruit of horticultural species [22,23,26,34]. We suspect that *CmARM14* may be the key regulatory gene for melon MFA. To understand whether *CmARM14* regulates fruit abscission, interaction proteins were predicted, and the results revealed that BRASSINOS-TEROID INSENSITIVE 1-associated receptor kinase 1-like isoform X1 (*BR1*, XP_008455226.1, XP_008440121.1), RING-type E3 ubiquitin transferase (XP_008446840.1), the protein kinase *PINOID2* (XP_008457762.1), chitin elicitor receptor kinase 1 (XP_008455461.1), and six other proteins that could interact with it (Figure S2). For BRASSINOSTEROID INSENSITIVE 1 (BRI1), the effect on ethylene biosynthesis and fruit ripening has been documented in horticultural plant species, including strawberry [35] and tomato [36]. More recently, Ji et al. (2021) reported a conserved mechanism by which brassinosteroids (BRs) suppress ethylene biosynthesis during climacteric fruit ripening [1].

How to finely regulate fruit abscission in horticulture plants to produce clear patterns is a fascinating but largely unresolved question [21]. Although our research primarily results in a map of a candidate gene for melon MFA, how the gene is regulated and the underlying molecular mechanism should be investigated.

**Supplementary Materials:** The following supporting information can be downloaded at: https://www.mdpi.com/article/10.3390/agronomy12123117/s1, Figure S1: Natural population sequence analysis and electropherograms. A. Sequence analysis of the candidate gene *CmARM14* in different varieties (2150–2200 bp). In 7 natural populations, S8 and S7 are MFA varieties. The other varieties are None MFA. B. Electrophoretic maps of the markers screened for detection in 7 natural populations. No.1,3 and 4 are M2-10. No.2 is ZT00091. No.5–14 are 1244, MS5, WI998, P5, P10, S8, and S7; Figure S2. Prediction of the interaction protein with *CmARM14*. Using https://cn.string-db.org/cgi (accessed on 2 June 2022) web to predict interacted protein with *CmARM14*; Table S1: MFA segregation of fruit mature abscission for F$_3$ families; Table S2: Candidate gene qRT-PCR primer and subcellular localization primer information.

**Author Contributions:** Y.S. designed this experiment and wrote the manuscript. D.D. performed the research and prepared figures and tables. L.W. and H.Q. collected phenotypic data in field trials. J.Z. conducted data analysis. H.L. gives specific modification opinions and constructive suggestions for submission. All authors have read and agreed to the published version of the manuscript.

**Funding:** This study was supported by the National Natural Science Foundation of the People's Republic of China (Nos. 31772330) and Natural Science Foundation of the Heilongjiang Province (Nos. LH2022C065) the Heilongjiang Bayi Agricultural University Support Program for San Zong (TDJH202004).

**Institutional Review Board Statement:** Not applicable.

**Informed Consent Statement:** Informed consent was obtained from all subjects involved in the study.

**Data Availability Statement:** The original contributions presented in the study are publicly available. This data can be found here: https://www.ncbi.nlm.nih.gov/bioproject/PRJNA908137 (accessed on 4 December 2022).

**Acknowledgments:** Not applicable.

**Conflicts of Interest:** The authors declare that they have no conflict of interest.

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
