# Peer review of "Primary Mapping and Analysis of the CmARM14 Candidate Gene for Mature Fruit Abscission in Melon"

_agronomy, doi:10.3390/agronomy12123117_

Round 1

Reviewer 1 Report

The research goal and obtained results seem to be original and valid. However, except the Abstract and Introduction, the rest of the manuscript requires serious improvement of English. Manuscript needs clarification (language improvements) of many parts stated in Material & Methods and Results chapters. Descriptions of some figures are not clear. I strongly suggest the manuscript to be checked by senior researcher once again.

A very good example how to do it is the paper of the same research group: Wang L, Li J, Yang F, Dai D, Li X and Sheng Y (2022) A preliminary mapping of QTL qsg5.1 controlling seed germination in melon (Cucumis melo L.). Front. Plant Sci. 13:925081. doi: 10.3389/fpls.2022.925081

 Some illustrations of pure editing and unclear statements:

Manuscript line (36) Introduction and (111) Material and Methods share the same chapter number - 1.

Manuscript line 125: For investigation the mature fruit abscission, each plants investigated the mature fruit from 25 days to 30 days after pollination. For families, fruit from all plants were detected, fruit were harvest and ripening behavior was classified, recording “A” as all plant fruit abscission, “B” as all plant fruit non-abscission, and fruit behavior abscission and non-abscission record as “H”.

Line 162: ...the candidate region. For the fine mapping candidate gene, totally 352 F3-57 individuals plants were grown and genotyped with F4-57 families (each family contains 30 plants ) by polymorphsiam markers to the refine position of the locus.

Line 446: Figure 1. performance of M2-10 and ZT-091, and their offspring generation of F3 families F3-57-2 and F3-57-92. each materials has three fruit which from 15 DAP (days after pollination), 20 DAP, and 25 DAP. For M2-10, fruit became mature about 32 days after pollination and without abscission until ripen. Fruit of ZT-091 became mature from 28 DAP but started to abscission from ~25days. F3-57-2 and F3-57-92 was the F3 family of NO.57 from M2-10 crossing with ZT-091, fruit of them were without mature abscission and with abscission, respectively.

Line 452: Figure 2. Mapping of candidate gene for MAF. A:Primarily mapping of mfa10.1 using SLAF-BSA (Specific-Locus Amplified Fragment)technology by F3-57 family, which a single gene segregation population for AL4 gene.

Author Response

Thank you for your letter and for the reviewers’ comments concerning our manuscript entitled “Primary mapping and analysis of the CmARM14 candidate gene for mature fruit abscission in melon” (Manuscript Number: agronomy-2023338). The comments are all valuable and very helpful for revising and improving our paper, as well as the important guiding significance to our researches. We have studied comments carefully and have made correction which we hope meet with approval. Revised portion are marked in the paper. The main corrections in the paper and the responds to the reviewer’s comments are as following:

Manuscript line (36) Introduction and (111) Material and Methods share the same chapter number - 1.

Reply: Thank you for your advice. We have changed the section numbers.

Manuscript line 125: For investigation the mature fruit abscission, each plants investigated the mature fruit from 25 days to 30 days after pollination. For families, fruit from all plants were detected, fruit were harvest and ripening behavior was classified, recording “A” as all plant fruit abscission, “B” as all plant fruit non-abscission, and fruit behavior abscission and non-abscission record as “H”.

Reply: Thank you for your advice. We have revised the sentence. (line 95: with “A” representing all plants whose mature fruit didi not undergo abscission, “B” representing all plants whose mature fruit underwent abscission, and “H” representing plants whose mature fruit showed both phenotypes.)

Line 162: ...the candidate region. For the fine mapping candidate gene, totally 352 F3-57 individuals plants were grown and genotyped with F4-57 families (each family contains 30 plants ) by polymorphsiam markers to the refine position of the locus.

Reply: Thank you for your advice. We have revised the sentence. (line 25:  Additionally, 352 F3-57 plants and the F4-57 families (each containing 30 plants) were used to fine mapping the candidate genes. Plants were also genotyped to determine the candidate position of the locus by polymorphic markers. )

Line 446: Figure 1. performance of M2-10 and ZT-091, and their offspring generation of F3 families F3-57-2 and F3-57-92. each materials has three fruit which from 15 DAP (days after pollination), 20 DAP, and 25 DAP. For M2-10, fruit became mature about 32 days after pollination and without abscission until ripen. Fruit of ZT-091 became mature from 28 DAP but started to abscission from ~25days. F3-57-2 and F3-57-92 was the F3 family of NO.57 from M2-10 crossing with ZT-091, fruit of them were without mature abscission and with abscission, respectively.

Reply: Thank you for your advice. We have revised Figure 1 and the note. (Figure 1. Technology roadmap phenotype of M2-10 and ZT00091, and their offspring generation of F3 families F3-57-2 and F3-57-92. each materials has three fruit which from 15 DAP (days after pollination), 20 DAP, and 25 DAP. For M2-10, fruit became mature about 32 days after pollination and without abscission until ripen. Fruit of ZT-091 became mature from 28 DAP but started to abscission from ~25days. F3-57-2 and F3-57-92 was the F3 family of NO.57 from M2-10 crossing with ZT00091, fruit of them were mature abscission and with mature non-abscission, respectively. )

Line 452: Figure 2. Mapping of candidate gene for MAF. A:Primarily mapping of mfa10.1 using SLAF-BSA (Specific-Locus Amplified Fragment)technology by F3-57 family, which a single gene segregation population for AL4 gene.

Reply: Thank you for your advice. We have revised Figure 2 and the note. (Figure 2. Mapping of candidate genes for MAF. A:Primarily mapping of mfa10.1 using SLAF-BSA technology by F3-57 family, which a single gene segregation population. B: narrow mapping of candidate region in chromosome 10, and more 14 molecular markers were used to narrow the candidate region by another more 352 F3-57 individuals plants. C: at last the candidate gene was located between SNP271 to SNP286 which spanned 35 Kb region that contains 6 candidate genes.)

Author Response

Dear Reviewer:

Thank you for your letter and for the reviewers’ comments concerning our manuscript entitled “Primary mapping and analysis of the CmARM14 candidate gene for mature fruit abscission in melon” (Manuscript Number: agronomy-2023338). The comments are all valuable and very helpful for revising and improving our paper, as well as the important guiding significance to our researches. We have studied comments carefully and have made correction which we hope meet with approval. Revised portion are marked in the paper. The main corrections in the paper and the responds to the reviewer’s comments are as following file.

Reviewer 3 Report

The purpose of this study was to identify one of the genes of influence for melon fruit abscission after ripening.  This trait is of agricultural importance as it influences when and how fruit can be harvested.  The authors had two strains of melon, one of which showed fruit abscission before ripening and one which showed fruit abscission after ripening.  They bred these varieties together to obtain F1, F2, F3 families and backcross progeny.  These families were analyzed independently as the segregation analysis was indicative of at least 2 genes of major influence.  Here the authors focused on mapping and using sequence analysis to identify the causative allele for one gene.  They mapped the locus to a region of chromosome 9 with 6 genes then used Q-RTPCR of melon pedicles to find one gene with an expression pattern that was logical for the phenotype.  This gene was an E3 ligase.  The authors used localization of a GFP-fusion protein in tobacco to check for protein location and found this fusion protein was present in the Golgi and nucleus.  Overall, this work has important findings that could be directly applied to melon breeding to obtain the desired fruit abscission trait.  However, the presentation of these data needs improvement. 

Major concerns

1.       The current manuscript needs to clarify that this current study a second avenue of investigation as their published study “Cytological Observation of Fruit Peduncle Abscission Zone and Preliminary Mapping of Mature Fruit Abscission AL3 gene in Melon.”  While this reviewer understands that large-scale studies often generate more than one set of publishable results, the authors need to be upfront with the overall experimental setup and prior findings.  The two studies are highly complementary as a breeder would likely track both mature fruit abscission related genes.  The 2022 publication included a very nice table of the parental, F1, F2 plant numbers and phenotypes.  These data need to be included in the current manuscript in a clear manner, perhaps as a graphical summary of the work with phenotype numbers. 

2.       There needs to be an explanation of the genetics and pattern of inheritance of the trait.  Here, a figure showing the mapping population structure would be very helpful.  To clarify, what was the phenotype of the F1 population and for what % of the plants?  The prior publication states 30 F1 plants showed no mature fruit abscission had 0 had mature fruit abscission.  So, these data are consistent with MFA being a recessive trait.  What do the authors mean by “two complementary genes?”  Here a quick Punnett square of the F2 (assuming the situation where parentals are unlinked AABB x aabb, giving F1 AaBb, and segregating F2) shows 9 ways to be A_B_ and 7 ways to be homozygous recessive for at least 1 of the two genes, is this the 9:7 ratio the authors were mentioning? 

3.       The overall writing need substantial revisions.  For example, the authors need to frame the introduction to fruit maturity and abscission in a more structured manner.  What is the desired sequence of events for crop production, first ripening then abscission?  The authors allude to fruit abscission occurring due to external factors, such as powdery mildew infection.  Is this true?  To clarify, for any previously identified QTLs or loci associated with ripening or MFA, how many have the genes identified?  Here, a table could be very helpful a way to summarize prior findings.  Also, the introduction paragraph (lines 85-106) is a better fit for the discussion, where the identity of the mapped gene is discussed.

4.       There needs to be better validation of the candidate gene.  This could be accomplished by performing genotype/phenotype testing of additional F2 families (as the second gene of importance is also known that could be included in the analysis).  Ideally, there would be molecular validation of candidate gene function, but this is a lot of work.  Out of curiosity, is there a genetic transformation system for this type of melon? 

5.       I am concerned that the fusion protein is not functional and that the localization to Golgi and nucleus is an artifact.  Is this type of protein known to be functional with a C-terminal GFP tag?  

6.       Figure 5 needs a major revision as the legend does not match the panel labels and the panels lack letter designations.  The lower left panel is labeled GFP-Man1-mCherry but the legend states it is DAPI staining.  Please revise the figure. 

Minor concerns

7.       There are some small but important changes that need to be made to the text.  For example, the abstract states that the “gene was expressed in the nucleus.”  Please change the wording to “fusion protein was localized to the nucleus” as that better matches the data being described.  Line 319/320 also describes the localization as “in the Golgi . . . of the nucleus” which implies the Golgi is located within the nucleus (this was likely a typo with "of" instead of "and").  Please re-word this sentence. 

8.       Please add enzyme names and sites to the table of markers used.

9.       Please add the name of the vector used for protein localization.  A vector map would be a useful supplemental figure as well.

Author Response

(The authors gave the same response as above.)

Round 2

Reviewer 2 Report

Authors have addressed most of the comments and made necessary corrections to the language and style of writing. However, there were many other comments which were not addressed appropriately. I recommend the authors to provide a point to point response to the comments so that it is easily to review the paper. 

Author Response

Dear Reviewer:

Thank you for your letter and for the reviewers’ comments concerning our manuscript entitled “Primary mapping and analysis of the CmARM14 candidate gene for mature fruit abscission in melon” (Manuscript Number: agronomy-2023338). The comments are all valuable and very helpful for revising and improving our paper, as well as the important guiding significance to our researches. We have studied comments carefully and have made correction which we hope meet with approval.

Reviewer 3 Report

Thank you for your revisions to the manuscript.  Two main concerns remain that were not satisfactorily addressed in the initial revision.  Please address these concerns through changes to the manuscript itself with new experiments and revised figures. 

1.       Target gene validation.  The target gene needs to be validated by genotyping of at least some of the phenotyped populations.  This study must include the previously identified locus to see how the genotyping and genetic predictions match the phenotyping. 

2.       Protein localization.  The protein localization data, as presented (from tobacco), are not valid for the conclusion reached.  It cannot be determined from the panels shown which one is the free GFP control and which is the GFP-fusion protein.  These data must both be presented in order to compare the free GFP to the GFP-fusion.  Also, there are no clearly labeled nuclei in these images, which makes them not useful for determining if the fusion is localized to the nucleus.  If better images are available these should be used instead.  If these are the best images then the experiment needs to be repeated.  Perhaps a z-stack would help.  DAPI staining is suggested to help with visualization of nuclei. 

Author Response

(The authors gave the same response as above.)
